# Polymorphisms in Human *IL4*, *IL10*, and *TNF* Genes Are Associated with an Increased Risk of Developing NSAID-Exacerbated Respiratory Disease

**DOI:** 10.3390/genes13040605

**Published:** 2022-03-28

**Authors:** María Luisa Reigada-Rivera, Catalina Sanz Lozano, Esther Moreno Rodilla, Asunción García-Sánchez, Virginia García-Solaesa, Félix Lorente Toledano, Ignacio Dávila González, María Isidoro-García

**Affiliations:** 1Department of Clinical Biochemistry, University Hospital of Salamanca, 37007 Salamanca, Spain; mlrivera4@hotmail.com (M.L.R.-R.); virginia.garcia.solaesa@navarra.es (V.G.-S.); misidoro@usal.es (M.I.-G.); 2Biomedical Research Institute of Salamanca IBSAL, 37007 Salamanca, Spain; emrodilla@usal.es (E.M.R.); chonela@usal.es (A.G.-S.); lorente@usal.es (F.L.T.); 3Department of Microbiology and Genetics, University of Salamanca, 37007 Salamanca, Spain; 4Department of Allergy, University Hospital of Salamanca, 37007 Salamanca, Spain; 5Department of Biomedical Sciences and Diagnostics, University of Salamanca, 37007 Salamanca, Spain; 6Department of Pediatrics, University Hospital of Salamanca, 37007 Salamanca, Spain; 7Department of Medicine, University of Salamanca, 37007 Salamanca, Spain

**Keywords:** NSAID hypersensitivity, asthma, nasal polyposis, polymorphism, NERD

## Abstract

Background: The role of genetics in non-steroidal anti-inflammatory drugs (NSAID) exacerbated respiratory disease (NERD) is unclear, with different candidates involved, such as *HLA* genes, genes related to leukotriene synthesis, and cytokine genes. This study aimed to determine possible associations between 22 polymorphisms in 13 cytokine genes. Methods: We included 195 patients (85 with NERD and 110 with respiratory disease who tolerate NSAIDs) and 156 controls (non-atopic individuals without a history of asthma, nasal polyposis (NP), or NSAID hypersensitivity). Genotyping was performed by sequence-specific primer polymerase chain reaction (PCR-SSP). Amplicons were analyzed by horizontal gel electrophoresis in 2% agarose. Results: Significant differences in allele and genotype frequency distributions were found in *TNF* (rs1800629), *IL4* (rs2243248 and rs2243250), and *IL10* (rs1800896, rs1800871, and rs1800872) genes in patients with NSAID hypersensitivity. In all cases, the minor allele and the heterozygous genotype were more prevalent in NERD. An association of *TNF* rs1800629 SNP with respiratory disease in NSAID-tolerant patients was also found. Conclusions: Retrospectively recorded, we found strong associations of NERD with polymorphisms in *IL4*, *IL10*, and *TNF* genes, suggesting that these genes could be involved in the inflammatory mechanisms underlying NERD.

## 1. Introduction

NSAID-exacerbated respiratory disease (NERD) is characterized by the development of respiratory symptoms (asthma and/or rhinitis) induced by NSAIDs in patients with respiratory disease (rhinitis or sinusitis, nasal polyposis, and/or asthma). Patients react to multiple NSAIDs with different chemical structures [1,2].

The prevalence of NSAID hypersensitivity in asthmatic patients varies according to the diagnostic method: if the diagnosis is exclusively focused on the clinical history, the prevalence figures range from 4 to 11% [3,4]; however, if the diagnosis is based on oral controlled challenge tests, the prevalence increases to 21% [1,5]. In patients with asthma and nasal polyposis (NP), it can reach up to 24% [1,6].

The pathogenesis of rhinitis and asthma induced by NSAIDs is complex. Today, it is accepted that the disease can, at least partially, be the result of an alteration in the metabolism of arachidonic acid due to the inhibition of the enzyme cyclooxygenase 1 (COX-1) and the consequent synthesis and release of an increased amount of cysteinyl leukotrienes from eosinophils and mast cells in the mucosa of the target organ [7]. Similarly, it has been reported that, in these patients, the density of cysteinyl leukotriene receptors is increased in inflammatory mucosal cells [8], and there is an alteration in the metabolism of prostaglandins (PG) [9,10,11].

Several genetic candidates have been associated with NERD, including but not limited to *HLA* alleles [12,13] and genes encoding for enzymes related to NO synthesis [14] and the arachidonic acid pathway [15,16,17,18], although with controversial results. In addition, the *transcription factor T-box* gene [16,19] or genes encoding for cytokines (*IL4* [16,20], *IL10* [21,22,23], *TGFB1* [24], and *TNF* [25,26]) have also been implicated. The aim of this study was to evaluate whether specific polymorphisms of cytokine genes were associated with NSAID hypersensitivity. Identifying these associations could help classify patients with a greater predisposition to develop the disease, leading to better clinical management.

## 2. Materials and Methods

### 2.1. Population Studied

In this study, 351 individuals (195 patients and 156 controls) were included. A statistical power analysis was initially applied to calculate the sample size, which was determined for a statistical power ≥ 80% and a significance level ≤ 0.05 (http://statpages.org/proppowr.html accessed on 27 February 2022). This analysis was repeated at the end of the study to confirm the statistical power of the results. All subjects provided previous written informed consent. The study was approved by the local Ethics Committee (Ref 2013/09/124). Asthma and rhinosinusitis with nasal polyps were diagnosed according to GEMA [27] and EPOS guidelines [28], respectively. NERD was suspected when patients manifested respiratory symptoms (dyspnea, wheezing, cough, nasal discharge, and nasal congestion) after the intake of two different NSAIDs and had a history of asthma and/or rhinosinusitis with nasal polyps. The diagnosis of NERD should be consistent with these data from the clinical history. Oral provocation testing with ASA is not recommended in patients whose clinical history is highly compatible with NERD, as a severe respiratory reaction can be induced [1,29]. However, an oral challenge test with the culprit drug was performed in patients with an equivocal history. In these cases, the appearance of nasal or bronchial symptoms and/or a decline in FEV1 ≥ 20% was considered diagnostic of NERD. 

Controls should meet the following criteria: no symptoms or previous history of asthma, rhinitis, nasal polyposis, or NSAID hypersensitivity and negative skin prick tests to a battery of common inhalant allergens adapted to our environment [30]. In all cases, a diagnostic protocol of NSAID hypersensitivity was carried out by medical specialists. Controls should also have a history of tolerance to NSAIDs.

All subjects underwent skin prick testing with a local standard battery of common aeroallergens following the recommendations of the Subcommittee on Allergen Standardization and Skin Tests of The European Academy of Allergy and Clinical Immunology [30]. In addition, peripheral blood was obtained by venipuncture. Total serum IgE levels were measured using a fluoroenzyme immunoassay (FEIA; ImmunoCAP, Thermo Fisher Scientific Inc., Waltham, MA, USA) following the manufacturer’s instructions.

### 2.2. Molecular Analysis

An automated system extracted DNA from 1 mL of whole blood. The extraction protocol was performed using MagnaPure Compact equipment following the manufacturer’s recommendations (Roche Applied Science, Mannheim, Germany). The amount and quality of extracted DNA were evaluated by determining the absorbance at 260 nm and 280 nm in a BioPhotometer spectrophotometer (Eppendorf AG, Hamburg, Germany).

Because of the importance of the genes encoding cytokines involved in the inflammatory response associated with hypersensitivity reactions to NSAID, 22 polymorphisms in 13 genes of Th1 and Th2 cytokines were included in this study (Table 1). A cytokine genotyping kit (Invitrogen, Waltham, MA, USA) was used for genotyping analysis. This system has been certified by the European Community regarding requirements for In Vitro Diagnostics 98/79/EC. It is based on the polymerase chain reaction system and uses sequence-specific primer polymerase chain reaction (PCR-SSP) to detect any SNP [31]. This system, also called the amplification-refractory mutation system (ARMS), detects any mutation involving a single base change. The system only amplifies DNA when the sample includes the target allele. At the 3’ end of these allele-specific oligonucleotides is the nucleotide complementary to one of the variants of the polymorphic site to be analyzed. This primer can only be efficiently extended when its 3’ end matches perfectly with the template DNA of the sample [31]. Two independent PCR reactions were performed to genotype each polymorphism, each using a different allele-specific primer and sharing a primer designed to bind to an invariant region. Therefore, two PCR reactions were designed that shared the same standard primer but differed in the specific primers for each allele: one primer will amplify only the normal allele, and the other primer will amplify only the mutated allele. Therefore, the obtained amplification reveals the presence of the variant sample detected by the allele-specific primer used.

The amplification was carried out using a 96-well thermocycler (T Professional 96 Thermocycler, Biometra^®^ GmbH, Göttingen, Germany) under the following conditions: initial denaturation at 94 °C for 4 min followed by 10 cycles of denaturation at 94 °C for 30 s and annealing and extension at 65 °C for 1 min; 20 cycles of denaturation at 94 °C for 30 s, annealing at 65 °C for 50 s, and extension at 72 °C for 1 min. Once PCR was completed, the products were analyzed by electrophoresis in a 2% agarose gel.

The electrophoresis analysis of the different samples yielded the corresponding allele and genotype frequencies for each SNP and the corresponding haplotype and diplotype frequencies. Two independent data analyses were performed to ensure the correct interpretation of results. In all steps of the experimental procedure, the recommendations of the European Molecular Genetics Quality Network (EMQN) [32] were followed, accessed on 27 February 2022. Appendix A shows an example of the electrophoresis results of one genotyping analysis.

### 2.3. Statistical Analysis

A descriptive analysis of the study population was performed using SPSS version 18.0 (Chicago, IL, USA). To detect any possible associations, a bivariate statistical analysis comparing the different allelic and genotypic distributions of the polymorphisms between patients and controls was performed. This was carried out using the online platform SHEsis (http://analysis.bio-x.cn/myAnalysis.php accessed on 27 February 2022) [33]. The chi-square test, Fisher’s exact test, and Monte Carlo (104 simulations) were performed for dichotomous variables; ANOVA was employed for continuous variables across each genotype prior to the analysis of homogeneity of variance. Hardy–Weinberg equilibrium was evaluated in all cases. Logistic regression was employed to model the effects of multiple covariates, including sex and age as potential covariates. Bonferroni correction for multiple comparisons, false-positive report probability (FPRP) using the method described by Wacholder et al. [34], and statistical power (SP) using the online platform http://statpages.org/proppowr.html accessed on 27 February 2022 [35] were also calculated.

Besides the analysis of single polymorphisms, the cytokine genotyping system allows the haplotype and diplotype analysis of specific genes, specifically *IL1B*, *IL2*, *IL4*, *IL6*, *IL10*, *TGFB*, and *TN**F*. This analysis was also performed with the statistical online platform SHEsis [33] (http://analysis.bio-x.cn/myAnalysis.php accessed on 27 February 2022) and with SPSS 18.0.

Therefore, the control of multiple associations found was carried out by multiple regressions, statistical power, and the false-positive report probability.

## 3. Results

### 3.1. Descriptive Analysis

Out of the 351 individuals included, 85 patients suffered from NERD: 83.5% (71) had both NP and asthma; 14.1% (12) had asthma but not NP, and 2.4% (2) had NP as the only respiratory disease. In addition, 110 patients with respiratory diseases (70 with NP and 40 with NP and asthma) that tolerated NSAIDs were included. The other 156 subjects were the control group. Table 2 presents the clinical and demographic characteristics of the different groups included in the study, including age and sex distributions. 

### 3.2. Analysis of Single Polymorphisms

#### 3.2.1. Patients with NSAID Hypersensitivity vs. Controls

When comparing SNP distributions in patients with NSAID hypersensitivity and controls, statistically significant differences were found in the following SNPs: *TNF* rs1800629, *IL4* (rs2243248 and rs2243250), and *IL10* (rs1800896, rs1800871, and rs1800872) (Table 3 and Table 4). The allelic and genotypic distributions of all SNPs included in the study in the NSAID hypersensitivity group and the statistical comparison with the control group are shown in Appendix A, respectively.

The A allele and the GA genotype of the *TNF* rs1800629 SNP were more frequent in NERD patients than in controls (*p* < 0.001); the statistical power for an α error of 0.05 was >99%, and the FPRP was 0% for an a priori probability of 0.1. This association was confirmed after adjusting for age and sex (*p* < 0.001; OR: 6.15; 95% CI (3.04–12.43)).

The TG genotype of the *IL4* rs2243248 SNP was more frequently observed in NERD patients than in controls (*p* < 0.001; OR: 2.07; 95% CI (1.09–3.933)). The SP was 94%, and the FPRP was 3.2%. This association was confirmed after adjusting for age and sex (*p* < 0.001).

Concerning the *IL4* rs2243250 SNP, a higher frequency of the T allele (*p* = 0.002) and the CT genotype (<0.001; OR: 0.45; 95% CI (0.28–0.74)) was observed in NERD patients. In the analysis of the genotype frequency, the SP was 90%, and the FPRP was 0%. Using logistic regression to adjust for age and sex confirmed this association *p* < 0.001.

Significant associations were found in genotypic frequencies of SNPs rs1800896, rs1800871, and rs1800872 in the *IL10* gene (Table 4). For rs1800871 and rs1800872, the CT and CA genotypes showed statistically significant differences (*p* = 0.006 and *p* = 0.008, respectively), showing higher frequencies in NERD patients than in controls. After adjusting for age and sex, a significant association was maintained for the two polymorphisms of *IL10* (*p* = 0.016 and 0.015, respectively).

#### 3.2.2. NSAID-Tolerant Patients vs. Controls

Comparing patients with respiratory diseases and NSAID tolerance and controls, a statistically significant association was only found in the *TNF* rs1800629 SNP (Table 3 and Table 4). The A allele and the GA genotype were more frequently found in patients than in controls (*p* = 0.003 and *p* = 0.004, respectively). In the case of the GA genotype, the SP was 92%, and the FPRP was 5.4% for an a priori probability of 0.1.

#### 3.2.3. Patients with NSAID Hypersensitivity vs. NSAID-Tolerant Patients

Statistically significant differences were also observed for the following SNPs: *TNF* rs1800629, *IL4* (rs2243248 and rs2243250), and *IL10* (rs1800896, rs1800871, rs1800872) between NSAID-tolerant and hypersensitive patients (Table 5). 

The mutated A allele and the GA genotype of the *TNF* rs1800629 SNP were more frequent in patients with NSAID hypersensitivity than in NSAID-tolerant patients (*p* = 0.005 and *p* = 0.015, respectively), although for both the A allele and GA genotype, the SP was less than 80%. This was confirmed after adjusting for age and sex (*p* = 0.036; OR: 0.50; 95% CI (0.25–0.99)).

The results for *IL4* (rs2243248 and rs2243250) and *IL10* (rs1800896, rs1800871, and rs1800872) SNPs are listed in Table 5. A higher frequency of the mutated allele and the heterozygous genotype was found in patients with NSAID hypersensitivity than in NSAID-tolerant patients for *IL4* SNPs. Regarding *IL10* SNPs, only the heterozygous genotype was associated with hypersensitivity to NSAID.

### 3.3. Haplotype and Diplotype Analysis

When performing the haplotype and diplotype analysis of *TNF*, *IL4*, and *IL10* genes, statistically significant associations were found in patients with NSAID hypersensitivity when compared to controls (Table 6). Significant differences were found for the same SNPs by comparing the haplotype and diplotype frequencies of patients with NSAID hypersensitivity and NSAID-tolerant patients (Table 7). In the haplotype and diplotype analysis of *TNF* rs1800629 and rs361525 SNPs, the AG haplotype and the AG GG diplotype were more frequent in patients with NSAID hypersensitivity. However, the GG haplotype and the GG GG diplotype were more frequently found in NSAID-tolerant patients (Table 7).

Regarding NSAID-tolerant patients compared with controls, a statistically significant association was found only in the haplotype and diplotype analysis of *TNF* and *IL4* (Table 6).

## 4. Discussion

In this study, we analyzed 22 polymorphisms corresponding to 13 genes in 351 individuals. First, we compared a population of NSAID hypersensitivity patients with controls. Secondly, we compared NSAID hypersensitivity patients and NSAID-tolerant patients, and finally, we compared a population of NSAID-tolerant patients with respiratory disease to controls. We found statistically significant differences between NSAID hypersensitivity patients and controls for SNPs of *IL4*, *IL10*, and *TNF* genes. These differences were maintained when NSAID hypersensitivity and NSAID-tolerant patients were compared. Interestingly, when tolerant patients were compared to controls, differences in the frequencies of *TNF* SNP were also observed. 

We found a statistically significant association of *IL4* rs2243248 and rs2243250 polymorphisms with hypersensitivity to NSAIDs in in the analysis of both allelic and genotypic frequencies. In the haplotype and diplotype analysis for the rs2243248/rs2243250/rs2070874 *IL4* SNPs, a statistically significant association was also found. Many published studies found an association between polymorphisms in the *IL4* gene and asthma [36,37,38]. Polymorphisms in *IL4* have also been associated with chronic rhinosinusitis [39] and nasal polyposis [39,40]. Nevertheless, there is scarce information about the influence of *IL4* polymorphisms in asthmatics with aspirin hypersensitivity (AIA). In a study in a Korean population [20], the frequency of the T allele of the *IL4* rs2243250 polymorphism was higher in the group with AIA than in the group of aspirin-tolerant asthmatics, which agrees with our results. A recent review by Gómez et al. [17] confirmed the data related to the *IL4* rs2243250 polymorphism. Aspirin and other NSAIDs could regulate the expression of the IL-4 allele, modifying the binding capacity of transcription factors, which could underlie the pathogenesis of hypersensitivity to aspirin or other NSAIDs [20,41].

Regarding the *IL10* gene, we found a strong association of three polymorphisms (rs1800896, rs1800871, and rs1800872) with hypersensitivity to NSAIDs. In all cases, the heterozygous genotype was more frequent in patients with NSAID hypersensitivity compared to NSAID-tolerant patients and controls. An association of ATA GCC and ACC GCC diplotypes with NSAID hypersensitivity was also found. Numerous published studies found an association between polymorphisms, chiefly located in the promoter region of *IL10*, and asthma [42,43,44] or COPD [45,46]. However, here, we focus on some studies that found an association between polymorphisms of *IL10* and aspirin sensitization in asthmatic patients [23] and patients with urticaria [22]. In the work of Kim et al. [23], the authors identified the G allele of *IL10* rs1800896 as a risk factor for aspirin hypersensitivity in asthmatics with rhinosinusitis. Moreover, they found that this effect would be enhanced by genetic interactions between the polymorphisms rs1800896 of *IL10* and rs1800469 of TGFβ1. Recently, an *IL10* polymorphism (rs1800872) was associated with NSAID-exacerbated respiratory disease in a Mexican mestizo population [21]. Our results seem to support the published evidence regarding heterozygous patients.

In the case of the *TNF* gene, the polymorphism rs1800629 showed a highly significant statistical association with patients with NSAID hypersensitivity. Interestingly, differences were also obtained in patients tolerant to NSAIDs compared to controls. These results suggest that this gene could be involved not in intolerance but in the inflammatory mechanisms underlying the respiratory disease. In the literature, numerous studies have found an association between polymorphisms of *TNF* with the susceptibility and severity of asthma, especially the polymorphism rs1800629 [47,48,49]. An association of *TNF* rs1800629 with nasal polyposis has also been found [50,51]. Regarding NSAID hypersensitivity, Kim et al. [26] established that polymorphisms in the promoter region of *TNF* significantly increased the susceptibility to aspirin-induced asthma (AIA) by the gene–gene interaction with the genetic marker *HLA DPB1* * 0301. This group suggested a possible linkage of SNPs in the promoter region of *TNF* with other nearby genetic markers that alter the expression levels of the cytokine, thus contributing to the development of aspirin hypersensitivity. In a subsequent review, a Hungarian group showed [25] that the *TNF* rs1800629 A allele was a genetic predisposition factor in a subgroup of patients with aspirin sensitivity, chronic rhinosinusitis (CRS), and nasal polyposis (NP). This group further suggested that a conserved linkage group on the short arm of chromosome 6 could be responsible for the genetic predisposition to this disease. These results were later confirmed by the same group [52]. In a recent study in a Mexico mestizo population, Pavón-Romero et al. found that the GA genotype of rs1800629 in *TNF* was associated with the risk of developing AERD (*p* < 0.05; odds ratio = 2.36) and by the dominant model (*p* < 0.05; odds ratio = 2.51). Furthermore, they also found a difference in TNF serum levels between the aspirin-tolerant asthmatic group and the AERD patient group (*p* < 0.001). In conclusion, they described that the GA genotype of rs1800629 was associated with genetic susceptibility to AERD, but it did not correlate with protein serum levels in that group [53]. These results align with those found in our study, where the highest frequency for the rs1800629 GA genotype was detected in NSAID hypersensitivity patients. However, it should be noted that in our population, aspirin-tolerant patients also showed a frequency of the GA genotype that was significantly higher than that in the control population.

Concerning the statistical power of the different associations detected, in general, we observed that the statistical power was in line with the *p*-value, so for significance values of around 0.001, the SP is usually around 90%. However, various factors can influence the statistical power: population heterogeneity, variation in the size of the subgroups, or the magnitude of the effect of interest.

## 5. Conclusions

We found strong associations of NERD with polymorphisms in *IL4*, *IL10*, and *TNF* genes, suggesting that these genes could be considered candidate genes in NERD. However, *TNF* seems to also be involved in respiratory disease in NSAID-tolerant patients, suggesting that it could be implicated in the inflammatory mechanism underlying the respiratory disease. Further studies in other populations and functional studies could confirm these associations.

## Figures and Tables

**Table 1 genes-13-00605-t001:** SNPs analyzed in the study.

Genes	NCBI SNP ID	SNP Change
*IL1A*	rs1800587	−889 C > T
*IL1Β*	rs16944	−511 C > T
	rs1143634	3954 C > T
*IL1R*	rs2234650	*pst*IC > T 1970
*IL1RA*	rs315952	*msp*AI T > C 11,100
*IL2*	rs2069762	−714 T > G
	rs2069763	114 G > T
*IL4*	rs2243248	−1098 T > G
	rs2243250	−589 C > T
	rs2070874	−33 C > T
*IL4RA*	rs1801275	Gln221Arg A > G
*IL6*	rs1800795	−174 G > C
	rs1800797	−597 G > A
*IL10*	rs1800872	−592 C > A
	rs1800871	−819 C > T
	rs1800896	−1082 A > G
*IL12B*	rs3212227	pos 1188 A > C
*IFN*	rs2430561	874 A > T
*TGFB1*	rs1982073	869 T > C
	rs1800471	915 G > C)
*TNF*	rs361525	−238 G > A
	rs1800629	−308 G > A

NCBI ID: National Center for Biotechnology Identification.

**Table 2 genes-13-00605-t002:** Descriptive analysis of patients with NSAID hypersensitivity, NSAID-tolerant patients, and controls.

				Fisher’s *p*-Value		
	CONTROLS (N = 156)	NH (N = 85)	NTP (N = 110)	NH vs. C	NTP vs. C	NH vs. NTP
AGE Median (IR)	50 (33)	49 (21)	48 (21)	0.09	0.61	0.20
SEX	38% M,	37% M,	68% M,	0.89	<0.001	<0.001
	62% F	63% F	32% F			
TOTAL IgE (kU/L)	91.62	206.43	147.14	<0.001	0.018	0.08
Mean (SD)	(±175.82)	(±232.5)	(±168.5)			
POLYPOSIS	-	85.9%	100%	-	-	<0.001
ASTHMA	-	97.6%	36.4%	-	-	<0.001
POSITIVE SKIN PRICK TESTS	-	30.6%	30.2%	-	-	0.95
MONOSENSITIZATION	-	42.3%	53.1%	-	-	0.55
POLYSENSITIZATION	-	57.7%	46.9%	-	-	0.55

C: controls; NH: NSAID hypersensitivity; NTP: NSAID-tolerant patients. N: number of patients; M: male; F: female. The data are represented by the mean ± standard deviation, although age data are presented by the median ± interquartile range (IR).

**Table 3 genes-13-00605-t003:** Allelic frequency distributions of SNPs showed a significant association in NSAID hypersensitivity and NSAID-tolerant patients compared with controls.

**rs1800629** ** *TNF* **	** *n* **	**G**	**A**	*** *p*-Value**	**OR; 95% CI ***	**SP**	**FPRP** **(10%)**
C	152	0.94	0.06				
NH	84	0.75	0.25	<0.001	5.03; (2.82–9.01)	98%	0%
NTP	109	0.86	0.14	0.004	2.39; (1.31–4.38)	60%	1.3%
**rs2243248** ** *IL4* **	** *n* **	**T**	**G**	*** *p*-Value**	**OR; 95% CI ***	**SP**	**FPRP** **(10%)**
C	139	0.94	0.06				
NH	80	0.86	0.14	0.023	2.07; (1.09–3.93)	43%	6.5%
NTP	107	0.94	0.06	0.69			
**rs2243250** ** *IL4* **	** *n* **	**C**	**T**	*** *p*-Value**	**OR; 95% CI ***	**SP**	**FPRP** **(10%)**
C	142	0.86	0.14				
NH	80	0.74	0.26	0.002	2.13; (1.31–3.47)	63%	7.1%
NTP	109	0.83	0.17	0.37			

* OR; 95% CI: odds ratio and 95% confidence interval. C: controls; NH: NSAID hypersensitivity; NTP: NSAID-tolerant patients. SP: statistical power; FPRP: false-positive report probability for a priori probability of 10%. * Fisher’s *p*-value for comparing each group of patients with the control group.

**Table 4 genes-13-00605-t004:** Genotypic frequency distributions of the SNPs showed a significant association in NSAID hypersensitivity and NSAID-tolerant patients compared with controls.

**rs1800629** ** *TNF* **	** *n* **	**GG**	**GA**	**AA**	*** *p*-Value**	**OR; 95% CI ***	**SP**	**FPRP** **(10%)**
C	152	0.89	0.10	0.01				
NH	84	0.56	0.38	0.06	<0.001	6.15; (3.04–12.43)	>99%	0%
NTP	109	0.73	0.26	0.01	0.003	3.73; (0.31–44.63)	92%	5.4%
**rs2243248** ** *IL4* **	** *n* **	**TT**	**TG**	**GG**	*** *p*-Value**	**OR; 95% CI ***	**SP**	**FPRP** **(10%)**
C	139	0.89	0.08	0.03				
NH	80	0.73	0.27	0.00	<0.001	7.33; (0.73–73.25)	94%	3.2%
NTP	107	0.89	0.11	0.00	0.26			
**rs2243250** ** *IL4* **	** *n* **	**CC**	**CT**	**TT**	*** *p*-Value**	**OR; 95% CI ***	**SP**	**FPRP** **(10%)**
C	142	0.75	0.21	0.04				
NH	80	0.51	0.47	0.02	<0.001	3.31; (1.82–6.02)	90%	0%
NTP	109	0.68	0.30	0.02	0.21			
**rs1800896** ** *IL10* **	** *n* **	**AA**	**AG**	**GG**	*** *p*-Value**	**OR; 95% CI ***	**SP**	**FPRP** **(10%)**
C	143	0.46	0.43	0.11				
NH	82	0.36	0.60	0.04	0.026	4.04; (1.11–14.69)	72%	2.9%
NTP	110	0.43	0.39	0.18	0.28			
**rs1800871** ** *IL10* **	** *n* **	**CC**	**CT**	**TT**	*** *p*-Value**	**OR; 95% CI ***	**SP**	**FPRP** **(10%)**
C	141	0.49	0.36	0.15				
NH	82	0.43	0.53	0.04	0.006	1.70; (0.96–3.02)	77%	5.9%
NTP	104	0.58	0.25	0.17	0.18			
**rs1800872** ** *IL10* **	** *n* **	**CC**	**CA**	**AA**	*** *p*-Value**	**OR; 95% CI ***	**SP**	**FPRP** **(10%)**
C	140	0.50	0.38	0.12				
NH	82	0.43	0.55	0.02	0.008	7.22; (1.58–32.94)	81%	2.8%
NTP	109	0.60	0.33	0.07	0.24			

* OR; 95% CI: odds ratio and 95% confidence interval. C: controls; NH: NSAID hypersensitivity; NTP: NSAID-tolerant patients. SP: statistical power; FPRP: false-positive report probability for a priori probability of 10%. * Fisher’s *p*-value for comparing each group of patients with the control group.

**Table 5 genes-13-00605-t005:** Comparative study of allelic and genotypic frequencies between patients with NSAID hypersensitivity and NSAID-tolerant patients.

SNP	Alleles and Genotypes	NH FREQ	NTP FREQ	* *p*-Value	OR; 95% CI *	SP and FPRP
**rs1800629** ** *TNF* **	A	0.25	0.14	0.005	2.09; (1.24–3.51)	47%, 1.4%
	GA	0.38	0.26	0.015	0.23; (0.03–2.08)	43%, 4.5%
**rs2243248** ** *IL4* **	G	0.14	0.06	0.006	2.68; (1.28–5.60)	44%, 2.3%
	TG	0.27	0.11	0.004	3; (1.38–6.52)	81%, 1.5%
**rs2243250** ** *IL4* **	T	0.26	0.17	0.033	0.58; (0.35–0.96)	60%, 8.4%
	CT	0.47	0.30	0.044	2.08; (1.14–3.80)	68%, 4.4%
**rs1800896** ** *IL10* **	G	0.33	0.38	0.40		
	AG	0.60	0.39	0.001	1.79; (0.97–3.30)	84%, 5.2%
**rs1800871** ** *IL10* **	T	0.31	0.30	0.89		
	CT	0.54	0.25	<0.001	2.90; (1.53–5.50)	99%, 0.3%
**rs1800872** ** *IL10* **	A	0.30	0.24	0.19		
	CA	0.55	0.33	0.007	5; (1.01–25.02)	87%, 4.3%

* OR; 95% CI: odds ratio and 95% confidence interval. NH: NSAID hypersensitivity; NTP: NSAID-tolerant patients; FREQ: frequency. SP: statistical power; FPRP: false-positive report probability for a priori probability of 10%. * Fisher’s *p*-value for the comparison between NH and NTP patients.

**Table 6 genes-13-00605-t006:** Haplotype and diplotype distribution frequencies of *TNF*, *IL4*, and *IL10* genes in NSAID hypersensitivity and NSAID-tolerant patients compared to controls.

			NH vs. C			NTP vs. C		
SNP	HAPLO	DIPLO	FREQ	* *p*-Value	SP	FREQ	* *p*-Value	SP
** *TNF* ** **rs1800629/rs361525**	AG GG	AG GG GG GG	0.25/0.06 0.32/0.09 0.67/0.86 0.46/0.75	<0.001 <0.001 <0.001 <0.001	98% >99% 92% >99%	0.13/0.06 0.22/0.09	0.009 0.004	51% 84%
** *IL4* ** **rs2243248/rs2243250/rs2070874**	TCC	TCC TCC	0.59/0.80 0.33/0.65	<0.001 <0.001	93% 99%			

FREQ: frequency. NH: NSAID hypersensitivity; NTP: NSAID-tolerant patients; C: controls. HAPLO: haplotype; DIPLO: diplotype. SP: statistical power. * Fisher’s *p*-value for comparing each group of patients with the control group.

**Table 7 genes-13-00605-t007:** Haplotype and diplotype distribution frequencies of *TNF*, *IL4,* and *IL10* genes in NSAID hypersensitivity compared to NSAID-tolerant patients.

			NH vs. NTP		
SNP	HAPLOTYPE	DIPLOTYPE	FREQ	* *p*-Value	SP
** *TNF* ** **rs1800629/rs361525**	AG		0.25/0.13	0.003	58%
		AG GG	0.32/0.21	0.021	36%
	GG		0.67/0.80	0.001	54%
		GG GG	0.46/0.66	0.021	80%
** *IL4* ** **rs2243248/rs2243250/rs2070874**	TCC		0.59/0.76	0.001	72%
		TCC TCC	0.33/0.59	0.005	96%
** *IL10* ** **rs1800896/rs1800871/rs1800872**		ATA GCC	0.37/0.20	<0.001	75%
		ACC GCC	0.45/0.15	<0.001	>99%
		GCC GCC	0.06/0.25	<0.001	96%

FREQ: frequency. NH: NSAID hypersensitivity; NTP: NSAID-tolerant patients. SP: statistical power. * Fisher’s *p*-value for comparing NSAID hypersensitivity and NSAID-tolerant patients.

## Data Availability

Data are contained within the article or Appendix A. Additional data presented in this study are available on request from the corresponding author.

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
