# Peer review of "Polymorphisms in Human IL4, IL10, and TNF Genes Are Associated with an Increased Risk of Developing NSAID-Exacerbated Respiratory Disease"

_genes, 2022, doi:10.3390/genes13040605_

Round 1

Reviewer 1 Report

From the abstract, some mistakes can be detected. Beginning with the incorrect use of trivial names for the analyzed SNPs/genes, studies trying to evaluate genetic/genomic association should be careful with the appropriate redaction symbols, this can be consulted from international and official databases, as HGNC (https://www.genenames.org/) to avoid the wrong use for common/trivial names. Also, this should be in higher-case and italicized letters as correspond to human genes.

The common or trivial nomenclature for SNPs (-308, -1098, -590, -1082, -819, and -592 etc.) is out of use, the authors should avoid it and correct it through the manuscript, only "rs" id is the right way to describe genetic variants. Also, the approved symbol for the tumor necrosis factor is TNF (without "A"), and in italics ALWAYS that you are referring to the gene (https://www.genenames.org/data/gene-symbol-report/#!/hgnc_id/HGNC:11892). Please correct throughout the entire manuscript.

Please move the study aim from the Material and methods section to the end to the introducción section.

Please explain how the genes and polymorphisms were selected in the current version only says: Because of the importance of the genes encoding cytokines in the inflammatory response associated with hypersensitivity reactions to NSAID…

Please describe the cytokine genotyping method adequately; the reference provided has more than 30 years, and "the Cytokine genotyping kit (InvitrogenTM, Deerbrook, Trail, USA)" appears discontinued in the supplier webpage. The authors should describe in deep the procedure.

Table 1 is not practical; I recommend moving to supplementary material; this should be corrected according to correct genes/SNPs nomenclature comments. In addition, in the “rs ID” the space between the “rs” and the number should be deleted.

Table 2 should be reformulated; the authors state: the data is represented by the mean ± standard deviation, but in the age line say “Medium (IR)” but the values correspond to SD; please correct, the appropriate is show IR; also correct Medium by Median.

In Table 3, the association is unclear if the-1098 IL4 is between NH/NTP vs. C. Please center to the correct line. Also, in the text for this same SNP (line 162-164), the TG genotype of the -1098 IL4 SNP, the OR, and 95% CI are not shown after adjusting by age and sex, please include these data accordingly, the same thing with the -590 IL4 SNP in the lines 165-168.

SNPs in the TNF gene and AERD/NERD has been previously studied in mestizo populations, particularly the GA genotype of rs1800629, which is associated in the current study; please discuss this previous study concerning your findings.

Finally, in the discussion section, authors should discuss the heterogeneity for statistical power.

Reviewer 2 Report

The article entitled “Polymorphisms in IL4, IL10, and TNFA Genes are Associated with an Increased Risk of Developing NSAID Exacerbated Respiratory Disease” by Reigada et al. aimed to determine the possible associations between 22 polymorphisms in 13 cytokines genes. Strong associations of NERD to polymorphisms in IL4, IL10, and 26 TNFA genes were found, suggesting that these genes could be involved in the inflammatory mechanisms underlying NERD. Functional studies could be done which the authors have mentioned as a limitation. The study is adding knowledge to the field and looks appropriate for the journal audience.

However, certain minor modifications are required.

Minor comments:

  1. Title - The highlight the importance of human studies, you could include human in the heading. For example, ‘Polymorphisms in human IL4, IL10, and TNFA Genes are Associated with an Increased Risk of Developing NSAID Exacerbated Respiratory Disease’ or so...
  2. Please use symbols of alpha and beta instead of A and B wherever TNFA and TGFB are used.
  3. It would be good if you could produce the electrophoresis images demonstrating gene polymorphisms.
  4. Table 3 – under -1098 IL4 gene - No. distribution is not shown. Please include the numbers.
  5. All values were seen adjusted by age and sex. Did you find any differences in the distribution of polymorphisms with respect to gender or age in addition to its association with cytokine gene polymorphism? Please clarify.
  6. Line 57 – two periods.

Round 2

Reviewer 1 Report

The authors have attended to my previous concerns. Please, just correct some minor mistakes in the redaction:

In the abstract, correct SSP-PCR by PCR-SSP and change the "mutated" by "minor" allele.
In Table 1, the column "Genes" still should be corrected the symbols for evaluated genes, mostly deleting the hyphen in the symbol, for example, IL1A instead IL-1A, etc.
Line 142, in the TNF symbol, the F is not in italics.
Table 2: The authors have corrected the median, and the abbreviation says "IR", but the values show SD (± 33, ± 21, etc.). According to the median, interquartile ranges should be displayed.
Table 3. In the second column, replace "No." with "n", too in Table 4.